# Respiratory Syncytial Virus Matrix Protein-Chromatin Association Is Key to Transcriptional Inhibition in Infected Cells

**DOI:** 10.3390/cells10102786

**Published:** 2021-10-18

**Authors:** Hong-Mei Li, Reena Ghildyal, Mengjie Hu, Kim C. Tran, Lora M. Starrs, John Mills, Michael N. Teng, David A. Jans

**Affiliations:** 1Department of Biochemistry and Molecular Biology, School of Biomedical Sciences, Monash University, Melbourne, VIC 3800, Australia; koubaihongmeili@icloud.com (H.-M.L.); Reena.Ghildyal@canberra.edu.au (R.G.); mengjie.hu000@gmail.com (M.H.); 2Centre for Research in Therapeutic Solutions, Faculty of Science and Technology, University of Canberra, Canberra, ACT 2617, Australia; Lora.Starrs@anu.edu.au; 3Department of Internal Medicine, Morsani College of Medicine, University of South Florida, Tampa, FL 33612, USA; KTran@health.usf.edu (K.C.T.); mteng@health.usf.edu (M.N.T.); 4Department of Infectious Diseases, School of Biomedical Sciences, Monash University and the Burnet Institute, Melbourne, VIC 3004, Australia; john.mills@monash.edu

**Keywords:** respiratory syncytial virus (RSV), RSV M protein, transcription inhibition, reverse genetics, nucleotide binding domain

## Abstract

The morbidity and mortality caused by the globally prevalent human respiratory pathogen respiratory syncytial virus (RSV) approaches that world-wide of influenza. We previously demonstrated that the RSV matrix (M) protein shuttles, in signal-dependent fashion, between host cell nucleus and cytoplasm, and that this trafficking is central to RSV replication and assembly. Here we analyze in detail the nuclear role of M for the first time using a range of novel approaches, including quantitative analysis of de novo cell transcription in situ in the presence or absence of RSV infection or M ectopic expression, as well as in situ DNA binding. We show that M, dependent on amino acids 110–183, inhibits host cell transcription in RSV-infected cells as well as cells transfected to express M, with a clear correlation between nuclear levels of M and the degree of transcriptional inhibition. Analysis of bacterially expressed M protein and derivatives thereof mutated in key residues within M’s RNA binding domain indicates that M can bind to DNA as well as RNA in a cell-free system. Parallel results for point-mutated M derivatives implicate arginine 170 and lysine 172, in contrast to other basic residues such as lysine 121 and 130, as critically important residues for inhibition of transcription and DNA binding both in situ and in vitro. Importantly, recombinant RSV carrying arginine 170/lysine 172 mutations shows attenuated infectivity in cultured cells and in an animal model, concomitant with altered inflammatory responses. These findings define an RSV M-chromatin interface critical for host transcriptional inhibition in infection, with important implications for anti-RSV therapeutic development.

## 1. Introduction

Human respiratory syncytial virus (RSV) is the main cause of acute viral lower respiratory tract infections (LRIs), particularly in infants <1 year old and the elderly [1], with the global annual burden of infection estimated at 64 million cases with 160,000 deaths [2]. Since there are currently no approved vaccines or generally available anti-viral drugs to combat RSV, detailed understanding of the molecular mechanisms underlying RSV disease is key to identifying new targets for antiviral development.

Like all negative-strand RNA viruses, RSV’s polymerase proteins are packaged in the virion, allowing transcription from and replication of the virus genome to proceed in the cytoplasm of the infected cell shortly after infection [3], followed by virion assembly [4]. Even though the RSV matrix (M) protein plays a key role in virion assembly at the plasma membrane [4,5,6,7,8,9,10], we have shown that it localizes in the nucleus of infected cells early in infection through the action of the conventional nuclear transporter importin β1 [11,12]. Significantly, nuclear extracts from RSV-infected cells early in infection are deficient in mediating transcription from an exogenous template, correlating with the period of nuclear localization of M [11], and implying that the nuclear function of M is to inhibit host cell transcription [5,11,13]. Later in infection, M traffics to the cytoplasm through the action of the host nuclear export protein CRM1 [14], where it inhibits viral transcriptase activity [7] through its RNA binding activity [15], and facilitates virus assembly by coordinating the activity of other RSV components [6,7,8,16].

The mechanism of inhibition of host transcription by nuclear M is unknown. The present study uses quantitative in situ analysis of host cell transcription and M-chromatin association, as well as in vitro approaches, to show that chromatin association of M is key to host transcriptional inhibition in RSV infection. Recombinant RSV with M impaired in chromatin association shows drastically decreased virus production in cell culture, and reduced infection in an animal model, concomitant with altered inflammatory responses. The results define an RSV M-chromatin interface critical for host transcriptional inhibition in infection, with important implications for anti-RSV therapeutic development.

## 2. Materials and Methods

### 2.1. Plasmid Constructs

The M cDNA was amplified from full-length RSV (A2 strain) cDNA [17]. Sequences encoding the M truncations (1–183, 63–183, 110–183, 1–110, 110–256, and 183–256) [12] were similarly amplified by PCR, while M-mutants (M-K121A, M-K130A, M-KK156/157TT, M R170A, M-K172A, M-R170T/K172T, M-R170A/K172TA) were amplified by overlap PCR. Amplified products flanked by attB sites were cloned into the pDONR 207 vector to generate Gateway™ entry clones (Invitrogen). Sequence fidelity was confirmed by sequencing (Micromon, Monash University). Entry clones were recombined with the destination vectors pDEST17 (to enable expression of hexa-His-tagged fusion protein in bacteria) and pEPI-GFP (to enable expression of GFP fusion protein in mammalian cells) [12,18]. Plasmid fidelity was confirmed by restriction analysis.

### 2.2. Cell and Virus Culture

RSV strain A2 was provided by Paul Young (University of Queensland, Australia). African green monkey kidney Vero, human cervical carcinoma HeLa, adenocarcinoma human alveolar epithelial cells A549 and human epithelial type 2 (HEp-2) cells were maintained in Dulbecco modified Eagle medium supplemented with 10% (vol/vol) fetal bovine serum (FBS), 50 U of penicillin/mL, 50 U of streptomycin/mL, and 2 mM l-glutamine. Virus stocks were grown in Vero cells as described [14] collected, and titres estimated by standard procedures [19].

### 2.3. Transfection

Overnight cultures of cells grown on glass coverslips were transfected using Lipofectamine 2000 (Invitrogen, Mulgrave, Australia) according to the manufacturer’s specifications. Cells were generally cultured for 24 h post-transfection prior to microscopic or other analysis.

### 2.4. CLSM Imaging and Image Analysis

Fixed and live (transfected or infected) cells were imaged using a Nikon Ti-E confocal laser scanning microscope (Nikon, Tokyo, Japan) in Kalman mode using a 100× oil immersion objective. The NIH ImageJ v1.62 public domain software was used as previously [14,20,21] to determine the nuclear/cytoplasmic fluorescence ratio (Fn/c), which was determined by using the equation: Fn/c = (Fn − Fb)/(Fc − Fb), where Fn is the nuclear fluorescence, Fc is the cytoplasmic fluorescence, and Fb is the background fluorescence (autofluorescence).

### 2.5. In Situ RNA Synthesis

The Click-iT RNA Imaging Kit (Invitrogen) was used according to the manufacturer’s specifications to assess de novo RNA synthesis in situ. Briefly, GFP/GFP fusion protein expressing Vero cells (23 h post-transfection) or RSV-infected cells (different time points post-infection) were treated with 1 mM EU for 1 h, fixed with 3.7% formaldehyde, permeabilized with 0.5% Triton X-100, and stained for de novo synthesized RNA using Alexa 594-azide, and for GFP using a specific monoclonal antibody to GFP and Alexa 488-coupled secondary antibody. Cells were then imaged by confocal laser scanning microscopy (CLSM).

### 2.6. Bacterial Expression and Purification of RSV M Fusion Proteins

*E. coli* BL21 cells carrying plasmid pDEST17-M (full length or point mutant derivatives) were grown at 37 °C in 400 mL LB containing 100 µg/mL ampicillin, followed by induction with IPTG (1 mM) for 4 h at 25 °C. The cell pellet was resuspended in 40 mL lysis buffer (50 mM Tris-HCl [pH 7.4], 500 mM sodium chloride, 25 mM imidazole) supplemented with 2 mg/mL lysozyme, Triton X-100 (0.1% *v*/*v* final concentration), 20 µg/mL DNase I, MgCl2 (1 mM final concentration), and Roche complete protease inhibitor cocktail, then incubated for 30 min on ice. The soluble fraction was isolated by centrifugation at 10,000× *g* for 30 min at 4 °C, and hexa-His-tagged proteins isolated using GE HisTrap FF 1 mL crude columns, following the manufacturer’s instructions, with proteins ultimately eluted using imidazole (up to 250 mM). Eluates were dialyzed overnight to remove the imidazole, and then concentrated using 10 K MWCO (Amicon Ultra) centrifugal concentrator devices.

### 2.7. In Vitro Transcription Assay

Various amounts of recombinant purified hexa-his tagged M protein (full length or mutant) were incubated with 100 pM of linearized template DNA (pGEM Express Positive Control Template) for 15 min at room temperature and then used in a transcription assay (T7 Riboprobe kit, Promega). Reactions (100 µL) were incubated for 1 h at 37 °C, and contained Transcription Optimized Buffer (1X), 10 mM DTT, 100 U of Recombinant RNasin Ribonuclease Inhibitor, 0.5 mM rNTP and 40 U of T7 RNA Polymerase) from the kit. Transcripts were subsequently analyzed by electrophoresis (1% agarose gel)/ethidium bromide staining, and the NIH ImageJ Imaging software used to determine the relative band intensities for the respective RNA products.

### 2.8. In Vitro DNA Binding Assay

Plasmid pEPI-Tag(110–135) [22] was used to examine the ability of M to bind DNA. 200 ng DNA was pre-incubated (15 min, room temperature) with increasing amounts (0–9.6 µM) of M protein in DNA binding buffer (10 mM HEPES, 0.25 mM DTT, 50 mM KCl) in a final volume of 10 µL, and then loaded onto a 0.8% agarose gel. After electrophoresis at 40 V for 8 h at 4 °C, the gel was stained with ethidium bromide solution before imaging using a GelDoc-It Imaging System (UVP). Band intensity was quantified by image analysis using the NIH ImageJ software, where the value for the non-shifted DNA band was calculated in percent relative to the total staining intensity of the lane.

### 2.9. In Vitro RNA Binding Assay

Analysis was essentially as for in vitro DNA binding (above) except that 200 ng in vitro transcribed RNA (c. 1800 bp control luciferase gene under control of the T7 promoter) was used. RNA was generated and precipitated as per manufacturer’s specifications using the RiboMAX Large Scale RNA Production System-T7 (Promega, Alexandria, Australia). Binding conditions, electrophoresis and imaging/quantitation were as for DNA binding (above).

### 2.10. Nuclear Association In Situ

Nuclear association of M was assessed in intact cells essentially as described [23]. In brief, overnight cultures of HeLa cells grown on coverslips were incubated for 15 min at 37 °C in 5% CO_2_ with Hoechst 33342 (Invitrogen) at 1 mg/mL dissolved in Dulbecco’s Modified Eagle Medium containing 10% FBS. Expression and subcellular localization of GFP fusion proteins was initially confirmed by live cell CLSM. Cells were then permeabilized with buffer A (10 mM Tris-HCl [pH 7.4], 150 mM NaCl, 5 mM MgCl2, 1% NP-40, protease inhibitor) for 15 min at room temperature and washed twice with buffer B (10 mM Tris-HCl [pH 7.4], 150 mM NaCl, 5 mM MgCl2, protease inhibitor) to remove the soluble proteins, and the subcellular localization of GFP fusion proteins visualized using CLSM. Cells were subsequently treated with buffer C (10 mM Tris-HCl, pH 7.4, 150 mM NaCl, 5 mM MgCl2, 0.2 mg/mL of DNase I or RNase 1 [Roche]/mL, protease inhibitor) for 1 h at room temperature. DNA-bound proteins were removed by incubating cells for 10 min at room temperature with buffer D (10 mM Tris-HCl, pH 7.4, 2 M NaCl, 5 mM MgCl2, protease inhibitor) followed by washing with buffer B; subcellular localization of GFP fusion proteins was subsequently visualized by CLSM.

### 2.11. Generation and Recovery of Mutant Recombinant RSV

Recombinant wild type RSV (rA2) was used as the template to derive the mutant recombinant RSV (rA2-M:R170T/K172T) as previously [14]. Briefly, the shuttle vectors pGEM-HX and pGEM-AX were constructed by cloning the HindIII-XhoI (positions 3280 to 4482) and AatII-XhoI (positions −98 to 4482) fragments of the full-length RSV antigenome (D53) [24] into pGEM-7Z(-). The HindIII-NcoI cassette of GFP-M:R170T/K172T containing the mutated residues was cloned into pGEM-HX and subcloned into pGEM-AX. Finally, the AatII-XhoI fragment of the mutant pGEM-AX constructs was cloned into D53. Presence of the desired mutation and absence of any other mutations was confirmed by sequencing. D53 plasmids were used for RSV recovery as described previously [25,26]. The rA2-M:R170T/K172T was extensively sequenced to confirm integrity of the M and other coding sequences (i.e., mutations were absent from the coding sequence for NS1).

### 2.12. Replication Kinetics of Recombinant RSV

Triplicate monolayer cultures of HEp-2, Vero or A549 cells in 6-well clusters were infected by the indicated viruses at various multiplicities of infection (MOIs). In each experiment, an aliquot of each inoculum was analyzed by plaque assay to confirm the input titer. Cell culture media from infected cells were collected at various time points and frozen in 50 mM HEPES pH 7.4/100 mM MgSO4. Virus titers were determined by plaque assay in Vero cells under 0.8% methylcellulose for 6 days. Plaques were visualized by immunostaining using a cocktail of three murine anti-F monoclonal antibodies followed by horseradish peroxidase-coupled anti-mouse IgG antibodies and 4CN substrate (Kirkegaard and Perry Laboratories, Gaithersburg, USA) as described [27] or by addition of 2.5 mL of neutral red diluted in 2% FCS/DMEM containing 1% SeaPlaque agarose; plaques were counted 18h later [28,29].

### 2.13. Infection and Immunofluorescence

Overnight cultures of Vero cells grown on glass coverslips were infected at an MOI of 3 with RSV strain A2, recombinant wild type (rA2), or M-R170A/K172A mutant or left uninfected (mock). Cells were fixed with 3.7% formaldehyde at 24 h post infection, followed by permeabilization with 0.5% Triton X-100. Fixed cells were immunostained with a monoclonal antibody specific for M (MAbαM) [30] and Alexa Fluor 488-conjugated secondary antibody (Invitrogen), mounted on slides using ProLong Gold followed by CLSM. RNA transcription in infected cells was studied using Click-iT RNA Imaging Kit as described above.

### 2.14. Real-Time RT-PCR to Quantify Viral RNA

Quantitative real-time polymerase chain reaction (RT-qPCR) was used to estimate the number of viral RNA genomes. RNA was extracted from cell-associated and cell-free virus using the Isolate II RNA kit (Bioline, Eveleigh, Australia), according to the manufacturer’s instructions. Nucleotide sequences (primers and probe) were directed to a region within the N gene of RSVA [31]. Primers (forward: 5′-CTC AAT TTC CTC ACT TCT CCA GTG T; reverse: 5′-CTT GAT TCC TCG GTG TAC CTC TGT) were synthesized by Integrated DNA Technologies. The probe (5′-TCC CAT TAT GCC TAG GCC AGC AGC A) was labeled with the 5′ reporter dye 6-carboxy-fluorescein (FAM) and the 3′ quencher dye 6-carboxy-tetramethylrhodamine (TAMRA) by Applied Biosystems. A one-step protocol was used with 10 μL of RNA added to each reaction mixture containing 1× TaqMan Fast Virus 1-Step Master Mix (Applied Biosystems, Scoresby, Australia), 300 nM of each of the primers, and 200 nM of the probe. The amplification profile used was 1 cycle for 5 min at 50 °C and 20 s at 95 °C, followed by 40 cycles for 3 and 30 s at 95 °C and 60 °C respectively. Absolute RNA copies were determined by extrapolation from a standard curve produced using a plasmid carrying the RSVA N gene cDNA [8].

### 2.15. Real-Time RT-PCR to Quantify Host Gene Expression

Expression of selected host genes was analyzed during infection by RT-qPCR. Total RNA was extracted from A549 cells infected without or with rA2 or rA2-M:R170T/K172T using the Isolate II RNA kit (Bioline, Eveleigh, Australia), according to the manufacturer’s instructions. cDNA was synthesized from 500 ng of total RNA using Superscript III First-Strand Synthesis System for RT-PCR (Invitrogen, Mulgrave, Australia) with Oligo(dt)20 primers. Quantitative PCRs for host genes were performed using SYBR green (Sensimix SYBR Hi-ROX, Bioline). Published gene-specific primer sets were used as follows: MCM5 (minichromosome maintenance deficient 5) [32], PSIP1/75 (PC4 and SFRS1 interacting protein 1 full length) [33], PSIP1/52 (PC4 and SFRS1 interacting protein 1 short isoform) forward primer [34], and reverse primer [35]. Tomm40 (translocase of outer mitochondrial membrane 40 homolog) [36], NDUFA10 (NADH dehydrogenase (ubiquinone) 1 alpha subcomplex, 10) [37]. IL12RB1 (interleukin 12 receptor, beta 1) [38]. PHB (prohibitin) [39]. SLC25A1 (solute carrier family 25 (mitochondrial carrier; citrate transporter), member 1) forward primer [40] and reverse primer 5′-AACCGTCTGCCGCACGCAGT-3′. TMEM11 (transmembrane protein 11) forward primer 5′-ACGACCTGCACAGAAAGAGA-3′; reverse primer 5′-ATACGGCATAGAGTTCGTAA-3′. MLF1IP (MLF1 interacting protein) forward primer 5′-TCCAGCCTTCCAGCTCTGTT-3′; reverse primer 5′-AGCTTCTCTAACTGATGGTTGA-3′. qPCR reaction conditions were as follows: 95 °C for 10 min, followed by 40 cycles of 30 s each of 95 °C, 60 °C, and 72 °C. Termination involved one cycle of 15 s each of 95 °C, 60 °C, and 95 °C. The copies of genes from rA2 and rA2-M:R170T/K172T infected cells were calculated relative to that of mock infected cells.

### 2.16. Animal Experiments

This study was performed in accordance with The ACT Animal Welfare Act (1992) and the Australian Code of Practice for the Care and use of Animals for Scientific Purposes, with the study protocol approved by the Committee for Ethics in Animal Experimentation of the University of Canberra (project reference number CEAE 14–15). Groups of 5 BALB/c mice (6–8 week old) were infected intranasally (i.n) as previously described [41] with 2.5 × 10^5^ pfu (in 50 µL) of rA2 or rA2-R170T/K172T; control mice received 50 µL of viral diluent. Mice were monitored daily for signs of disease (lethargy, ruffled fur) and weight loss. On days 3, 4, 5, and 7, mice were killed using cervical dislocation. Blood was collected by heart puncture, serum extracted and used for ELISA to determine systemic inflammatory responses. One lung from each mouse was fixed in formaldehyde for histology, and the other lysed in viral diluent with grinding beads using a TissueLyser II (Qiagen, Chadstone, Australia), for determination of pfu/lung.

Paraffin-embedded lungs were sectioned and processed for haemotoxylin and eosin (H&E) staining (Imaging & Cytometry Facility, John Curtin School of Medical Research, Australian National University, Canberra, Australia). Histological analysis of H&E-stained slides was used to determine perivascular vascular inflammation (PVI) and bronchial inflammation using quantification schema modified from that of others [42,43]; briefly, the intensity of perivascular or bronchial inflammation was scored numerically for each view-field on a scale of 1 to 9. 0 denotes no inflammation; 1–3, scant cells but not forming a defined layer; 4–6, 1–3 layers of cells surrounding the vessel; and 7–9, 4 or more layers of cells surrounding the vessel.

Systemic inflammation was estimated from serum samples using IL-6 and RANTES as markers, as per the manufacturers’ recommendations (IL-6, ELISAKIT.com, Melbourne, Australia; RANTES, RayBiotech Inc, Peachtree Corners, USA).

## 3. Results

### 3.1. RSV M Inhibition of Host Cell Transcription Parallels the Extent of Nuclear Accumulation

We previously showed that M localizes to the nucleus of RSV-infected cells early in infection [12]. To test the extent to which nuclear M contributes to inhibition of host cell transcriptional activity in nuclear extracts from RSV-infected cells [11], we set out to analyze de novo host cell nuclear RNA transcription in situ in RSV-infected Vero cells for the first time using labeling with EU (5-ethynyluridine) using the Click-iT RNA system [44] in combination with quantitative imaging by CLSM. M localization was assessed in parallel by immunostaining [7,8,45,46]. M was observed in both the cytoplasm and nucleus as previously (Figure 1A), but strikingly, infected cells showing M nuclear localization had markedly reduced EU incorporation compared to uninfected cells, indicative of inhibition of de novo nuclear RNA synthesis. This idea was supported by quantitative analysis at the single cell level (Figure 1B), results revealing that higher levels of nuclear localization correlated strongly with reduced levels of EU incorporation (see Figure 1C), clearly implying that nuclear M contributes to inhibition of host cell transcription in RSV infection.

To confirm that M was responsible for transcriptional inhibition, EU labeling was performed in cells transfected to express full length wild type M (Mwt) fused C-terminal to GFP (GFP-Mwt) or GFP alone (Figure 1D–F). GFP-Mwt expressing cells showed markedly reduced EU incorporation compared to cells expressing GFP alone (Figure 1D), consistent with the idea that RSV M is primarily responsible for the transcriptional inhibition observed in RSV-infected cells. Quantitative analysis (Figure 1E,F) further supported the idea that the extent of nuclear localization of M is a key determinant in transcriptional inhibition, with clear correlation between the two.

### 3.2. Arginine 170 and Lysine 172 within the Central Domain of M Are Critical for Transcriptional Inhibition

To determine which domains of M are responsible for transcriptional inhibition, truncation derivatives of M with or without the central RNA-binding region (amino acids 110–183—see Figure 2A, left), fused to GFP, were analyzed for the ability to reduce EU incorporation. The GFP-M-1-183 and -63-183 truncation derivatives were observed to inhibit transcription to the same extent as GFP-Mwt, whereas GFP-M-1-110 and -183-256 showed essentially no inhibition. By comparison, the GFP-M-110-183 and -110-256 truncation derivatives appeared to retain intermediate activity (Figure 2A right). Quantitative determination of the extent of de novo RNA synthesis using EU labeling confirmed these findings (Figure 2B), supporting the conclusion that the central domain of M (residues 110–183) is essential for inhibition of transcription (Figure 2C).

Rodriguez et al. [15] previously showed that M binds RNA, with the key residues being lysines 121, 130, 156, and 157 and arginine 170 within M’s central domain. We generated GFP-M point mutant derivative expression constructs for all of these (see Figure 3A top) in order to assess their ability compared to GFP-Mwt to decrease EU incorporation in transfected cells. Expression of GFP-M:K121A, K130A, or K156T/K157T all resulted in markedly reduced EU incorporation, in essentially identical fashion to GFP-Mwt (Figure 3A), quantitative analysis confirming that substitution of lysines 121, 130, 156, and 157 does not significantly impact this activity. Substitution of arginine at position 170, however, resulted in a 30% decrease in inhibition of EU labeling (Figure 3B,C). Alanine substitution of lysine 172, which is not involved in RNA binding, similarly effected a 25% decrease in transcriptional inhibition (Figure 3B,C). The double mutant derivatives (R170A/K172A or R170T/K172T) showed complete loss of inhibitory activity, EU labeling of de novo synthesized RNA in cells expressing them being identical to that of non-transfected cells (Figure 3A), as confirmed by quantitative analysis (Figure 3B,C). Similar results were obtained using bacterially expressed M proteins and the cell-free Riboprobe System (see Appendix A), where the R170T/K172T mutant derivative showed a c. 4-fold lower Ki than Mwt, the clear implication being that the R170/K172 residues are also essential for M’s transcriptional inhibition activity in vitro. Further, the microscopic analyses indicated a lack of an effect of the mutation of R170/K172 on M nuclear trafficking, with no reduction in nuclear accumulation or sensitivity to the nuclear export inhibitor leptomycin B evident (Appendix A). Residues arginine 170/lysine 172 are thus critical for inhibition of host cell transcription by M, independent of effects on RNA binding and nuclear translocation.

### 3.3. The R170T/K172T Mutant of M Binds RNA Normally, but Is Deficient in Transcriptional Inhibition and DNA Binding

Since R170 has previously been implicated in RNA binding [15], we set out to test whether nucleic acid binding was critical for M’s transcriptional inhibition activity. We first compared the ability of bacterially expressed Mwt and M:R170T/K172T mutant M proteins (of comparable concentration/purity—Figure 4B, right) to bind to a fixed amount of RNA as assessed in electrophoretic mobility shift assays. Results (Appendix A; not shown) indicated comparable RNA binding by both proteins (apparent association constants, Kas, of 2.1 and 3.6 µM, respectively, for Mwt and mutant proteins), implying that the R170T/K172T mutant is not significantly impaired in RNA binding. Since the R170T/K172T mutant clearly has reduced ability to inhibit transcription (Figure 3), however, this further confirms that RNA binding by M is not essential to transcriptional inhibition by M (see also above).

We next tested the ability of M to bind DNA, whereby increasing concentrations of wild type and mutant M were incubated with a fixed amount of plasmid DNA, and binding assessed by gel electrophoresis. DNA binding by Mwt was clearly evident at even 1.2 µM protein, with essentially complete retardation of the bands at 2.4 µM (Figure 4A); 1.6 µM was estimated to be the apparent association constant (Ka) (Figure 4B). In contrast, the M:R170T/K172T mutant showed markedly reduced DNA binding, with unshifted plasmid DNA observable even at the highest protein concentration of 9.6 µM (Figure 4A), and c. 4-fold reduced DNA binding affinity (Figure 4B; Ka of c. 6 µM). The results thus paralleled the difference in potency (see above) in terms of in vitro activity for transcriptional inhibition (Appendix A), suggesting that M is able to bind DNA with high affinity, with strong dependence on residues R170/K172. That the R170T/K172T mutant shows reduced DNA binding concomitant with impaired transcriptional inhibition implies that DNA binding by M may be necessary for inhibition of transcription by M.

To analyze M nuclear association in a cellular context, we applied an in situ visualisation approach that has previously been used to examine protein association with chromatin [23,47]. Briefly, we expressed GFP-Mwt or mutant derivatives thereof in HeLa cells and analyzed their subcellular localization after NP-40-permeabilization, whereby only proteins that associate with chromatin/the nuclear matrix are retained in the nucleus (Figure 5). As controls, we used cells expressing GFP itself, which does not bind to nuclear components, and GFP-histone H3, which binds tightly to cellular chromatin, as a positive control. GFP was not detectable after NP-40 permeabilization, as expected, whereas GFP-H3 was retained strongly in the cell nucleus (Figure 5A). Strikingly, GFP-Mwt was retained in the nucleus after NP-40 permeabilization, in contrast to essentially no retention observed for GFP, GFP-M:R170T/K172T, or R170A/K172A, and only low-level retention of GFP-M:R170A. These results correlated closely with those for in vitro DNA-binding (Figure 4 and not shown), and transcriptional inhibition in situ (Figure 3) and in vitro (Appendix A). Quantitative analysis confirmed these findings (Figure 5B,C), indicating significantly (*p* < 0.0001) c. 10-fold reduced nuclear association on the part of the M-R170T/K172T and R170A/K172A mutant derivatives compared to Mwt, supporting the conclusion that M protein associates with nuclear DNA dependent on residues R170/K172. That in situ nuclear association by M in this assay is likely to be due to DNA binding was indicated by the fact that DNase but not RNase treatment reduced nuclear association of M by >60%, in completely comparable fashion to results for H3 (Figure 5D–F). The clear implication is that DNA binding by M is central to transcriptional inhibition.

### 3.4. RSV Carrying Mutations in M Arginine 170/Lysine 172 Is Attenuated, Showing Concomitant Transcriptional Inhibition and Virus Production

To establish the relevance of the above results to RSV infection, we used the RSV reverse genetics system as previously [14,24,27,41,48] to generate a recombinant virus carrying mutations of arginine 170 and lysine 172. The mutant virus (rA2-M:R170T/K172T) showed a striking reduction in virus production in both Vero and HEp-2 cells compared to wild type recombinant rA2 virus (Figure 6A). In particular, there was a lag of about 24 h in virus production for rA2-M:R170T/K172T compared to rA2 in both cell lines; the effect was more pronounced in HEp-2, where 7 days post-infection (p.i.) the mutant virus titre remained 100 times lower than that of wild type. Similarly, significantly reduced titres of the mutant virus compared to wild type, were also observed in single cycle replication kinetics, where Vero or A549 cells were infected at an MOI of 3 and total virus titre determined by plaque assay at indicated times up to 48h p.i. (Figure 6B). That RSV with an M protein with greatly reduced DNA binding ability is markedly impaired in virus production clearly implies that DNA binding on the part of M is critical to RSV infection. Analysis of viral replication by RT-qPCR confirmed a > 8-fold difference between replication of rA2 compared to that of rA2-M:R170T/K172T at 24 h, implying that the mutant virus was deficient in early replication (Appendix A); replication in wild type and mutant virus infected cells was similar at 48 h.

To confirm that the impaired replication of rA2-M:R170T/K172T compared to wild type virus was due to reduced inhibition of host cell transcription, Vero cells were infected with rA2 or rA2-M:R170T/K172T at MOI of 3 followed by incubation with 1 mM EU for 1 h at 18 or 24 h post-infection. Cells were then fixed, permeabilized, and specifically stained for de novo synthesized RNA as well as M as per Figure 1. Although M could be readily detected in the nucleus in rA2-M:R170T/K172T-infected cells as early as 18 h p.i. (Figure 6C), host transcriptional inhibition was essentially unaffected, even up to 48 h p.i. (Figure 6D,E). This was in stark contrast to cells infected with wild type rA2, which showed significantly reduced host cell transcription at both 18 and 24 h p.i. (*p* < 0.0001). These results were confirmed for a set of specific genes based on previous studies [49], including anti-viral response genes such as IL12 receptor B1 and the protease prohibitin, and genes encoding mitochondrial proteins (see Section 2.15) [50,51,52], with clear effects for inhibition by wild type rA2, but not by rA2-M:R170T/K172T (Figure 6F) at MOIs of 1 or 3, and at various times 8–24 h p.i.

Taken together, these results are consistent with a critical role for M in the nucleus early in RSV infection that is dependent on M association with the host chromatin to effect transcriptional inhibition.

### 3.5. RSV Carrying Mutations in M Arginine 170/Lysine 172 Shows Reduced Viremia and Increased Anti-Viral Response in an Animal Model

To test the in vivo role of M chromatin association/transcriptional inhibition, we used the established BALB/c mouse model of RSV infection. Briefly, groups of 5 mice were infected intranasally (i.n) with rA2 or rA2-R170T/K172T while control mice received viral diluent. Consistent with previous studies in this model [24], intranasal inoculation with RSV resulted in decreased activity at day 2–3; mice inoculated with the mutant virus appeared to drink more water compared to those inoculated with rA2, but no significant changes in weight were observed compared to uninfected mice inoculated with viral diluent. Peak virus titres for both wild type and rA2-R170T/K172T viruses were reached at 4 days p.i., with gradual reduction on days 5 and 7 p.i., but the titre of the mutant virus was consistently lower (>4-fold lower at days 5 and 7, *p* < 0.05; Figure 7A). Histological analysis and scoring of lungs were performed as described [42,43,53], revealing significantly (*p* < 0.001) more inflammatory cell infiltrate around airways at days 3 and 4 p.i. in the mutant virus infected samples compared to inoculation with rA2 (Figure 7B,C). Significantly (*p* < 0.05) more systemic RANTES was observed in mice infected with wild type virus on days 5 and 7 p.i, compared to those inoculated with the mutant virus (Figure 7D). Levels of IL-6 were below the detection limit of the assay in all samples tested (not shown). The clear implication was that compared to wild type infection, infection with the mutant virus impaired in transcriptional inhibition resulted in reduced systemic inflammation at days 5 and 7 p.i., correlating with strongly reduced viral load.

## 4. Discussion

This study establishes that a key mechanism by which RSV inhibits transcription in the infected cell is through RSV M association with the host chromatin in the nucleus, dependent on specific residues in M’s central domain, representing the first report of host chromatin association for a pneumovirus M protein. Importantly, it provides the mechanistic framework to explain previous key observations relating to host cell transcription and mitochondrial function. Firstly, M:chromatin association would appear to be responsible, at least in part, for the global impact of RSV infection on the mRNA transcript levels of >3300 host genes [49,51], and concomitant large-scale changes in the host cell proteome [54,55]. Secondly, the resultant reduction in mRNA levels for mitochondrial genes would appear to be the basis of impaired mitochondrial function early in RSV infection [50,56]. Importantly, the results in Figure 6F reveal that RSV’s effects on the mitochondria are likely through RSV M-dependent down-regulation of mitochondrial gene expression that impacts the infected cell mitoproteome [54,55], and mitochondrial function in turn [50,56]. Figure 8 is a schematic model summarizing the details of this novel mechanism. RSV M nuclear trafficking/chromatin association leads to inhibition of host cell transcription, and of nuclear-encoded mitochondrial genes in particular, with downstream impact on mitochondrial function that favors infectious virus production [50,56].

Intriguingly, the role of M in this context resembles in part that of the bacteriophage λCro repressor, which acts to turn off early gene expression during the virus lytic cycle [57,58,59,60]. RSV M, like Cro, inhibits viral gene transcription, but in the case of RSV, this embodies the switch to initiate virus assembly. M-dependent inhibition of viral mRNA transcription too early in infection would halt the infectious cycle [7] in similar fashion to λlysogeny. Instead, M appears to be redirected to the nucleus early in infection to enable RSV transcription to occur in the cytoplasm, but importantly, also to allow M to moonlight as a potent inhibitor of host gene transcription important for the host antiviral response, including of nuclear-encoded mitochondrial genes. Although M proteins of other viruses, including paramyxoviruses such as Nipah virus, have been reported to localize in the nucleus of infected cells [61,62,63,64], RSV M may well be unique in terms of its ability to associate with the host cell chromatin and thereby impact transcriptional outcomes as shown in Figure 8.

This is the first study to use high-resolution, quantitative, in situ imaging based on state-of-the-art labeling to document M’s ability to inhibit de novo RNA synthesis in RSV-infected cells (Figure 1, Figure 2 and Figure 3 and Figure 6) and associate with nuclear chromatin (Figure 5). Importantly, our results derived using single cell analyses correlate perfectly with those from cell-free experiments with respect to both transcriptional inhibition and DNA binding, implying that RSV M’s ability to inhibit host cell transcription is strongly dependent on DNA binding, through the central region of the protein, and residues R170/K172 in particular. Mutation of these residues impairs transcriptional inhibition by M in both infected and ectopic expression systems (Figure 3 and Figure 6), reduces DNA but not RNA binding in situ (Figure 5) and in cell-free systems (Figure 4), results in an attenuated virus that is replication deficient compared to wild type in cultured cells (Figure 6), and induces a less robust infection in vivo (Figure 7), in parallel with enhanced early inflammation. Significantly, nuclear targeting, chromatin association, and transcriptional inhibition by M all appear to represent central aspects of RSV pathogenesis, and thereby targets for the development of urgently needed anti-viral agents to combat RSV. This is the main focus of future work in this laboratory.

## Figures and Tables

**Figure 1 cells-10-02786-f001:**
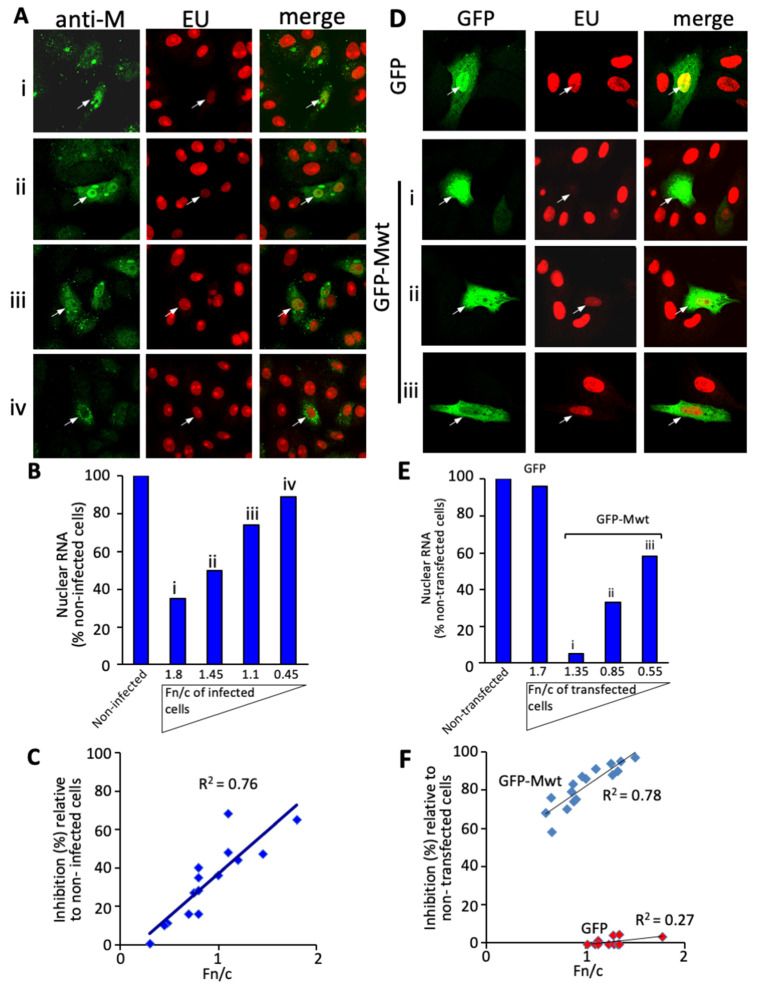
Inhibition of host transcription in RSV-infected cells is dependent on the extent of nuclear accumulation of RSV M. (**A**) Vero cells were infected with RSV at MOI of 3. At 24 h post-infection (p.i.), cells were incubated with 1 mM EU (see Methods) for 1 h, and then fixed, permeabilized, and specifically stained for de novo synthesized RNA using Alexa 594-azide, as well as for M using a specific monoclonal antibody followed by an Alexa 488-coupled secondary antibody. Representative CLSM images of subcellular localisation of M and de novo RNA synthesis in RSV infected cells derived using a 100X oil immersion objective (see Section 2.4) are shown. Merged images of the green (M protein) and red (EU incorporation reflecting de novo RNA synthesis) channels are shown; rows i-iv show cells, denoted by the arrows, with differing levels of intranuclear M, for which quantitative analysis is shown in (**B**). (**B**) Quantitation of the levels of M accumulation in the nucleus and EU incorporation from (**A**) by image analysis of the CLSM files in panel (**A**) using the NIH ImageJ software to determine the extent of nuclear accumulation of M (expressed in terms of the nuclear to cytoplasmic ratio, Fn/c) and levels of de novo synthesized RNA in infected cells (i–iv), indicated by arrows in (**A**), expressed as a percent of mock-infected cells in the same CLSM image in (**A**). (**C**) Linear regression (*n* = 15; correlation coefficient indicated) for the results for single cell analysis as per (**B**) for a series of images as per (**A**) (including all of the cells in **A**i–iv) for the relative amount of intranuclear M and the degree of transcriptional inhibition. (**D**) Vero cells transfected to express GFP or GFP-Mwt were treated with 1 mM EU 23 h after transfection for 1 h prior to fixation, permeabilization and staining for de novo synthesized RNA and M subcellular localization as above. Representative CLSM images of the green (GFP-Mwt or GFP alone) and red (EU incorporation) channels are shown, with merged images shown on the right. Rows i-iii for GFP-Mwt show cells, denoted by the arrows, with differing levels of intranuclear GFP-Mwt, for which the quantitative analysis is shown in (**E**). (**E**) Results for de novo RNA synthesis performed as in B, expressed as a percent of mock-infected cells in the same CLSM image in (**D**). (**F**) Linear regression (*n* = 16 and 10 for GFP-Mwt and GFP alone, respectively; correlation coefficients indicated) for the results for single cell analysis as per (**B**) for a series of images such as those in (**D**) for the relative amount of intranuclear GFP-Mwt and the degree of transcriptional inhibition.

**Figure 2 cells-10-02786-f002:**
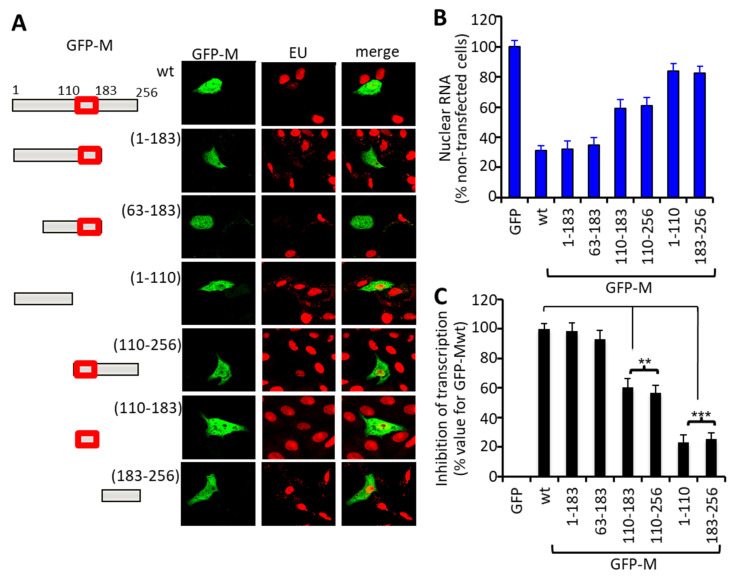
The central nucleotide binding region of RSV M is sufficient to effect transcriptional inhibition in living cells. (**A**) Schematic representations of the GFP-M fusion constructs used (left). Numbers refer to the M amino acid sequence, with the central domain, including the RNA binding region and nuclear localization signal (NLS), shown as a red box. Representative CLSM images of Vero cells expressing GFP-Mwt or truncated derivatives thereof (right) treated with EU, fixed, permeabilized, and stained for de novo synthesized RNA as per the legend to Figure 1. (**B**) Quantitative analysis for inhibition of de novo RNA synthesis (*n* = 15) as per the legend to Figure 1. (**C**) Results from (**B**) are expressed in terms of percent inhibition of transcription relative to that effected by GFP-Mwt. Significant differences are indicated, ** *p* < 0.01, *** *p* < 0.001.

**Figure 3 cells-10-02786-f003:**
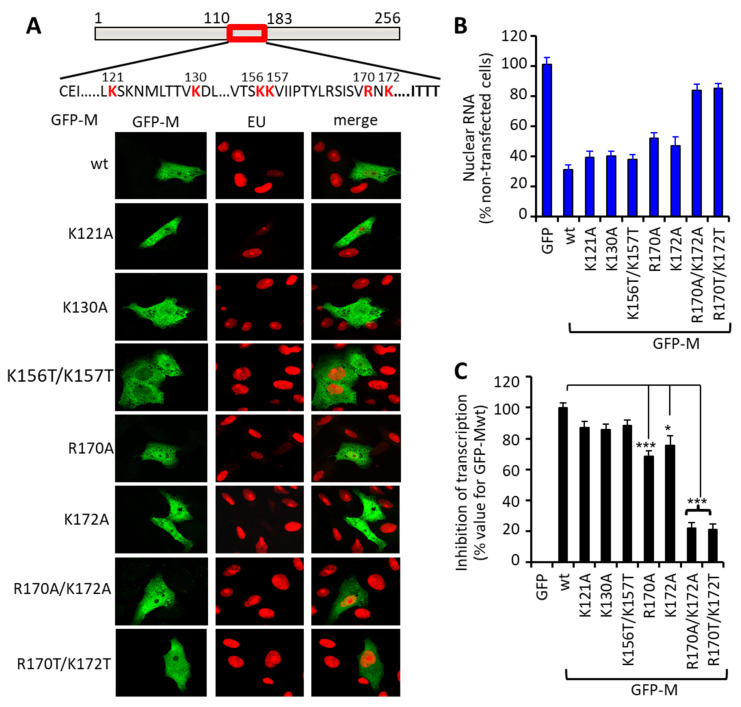
RSV M residues R170 and K172 within the nucleotide binding domain are critical for transcriptional inhibition in living cells. (**A**) Schematic representation as per Figure 2A of the sites of point mutation (red) within the central nucleotide binding domain of M, within the GFP-M fusion constructs used. Representative CLSM images of Vero cells expressing the indicated GFP-M constructs treated with EU, fixed, permeabilized, and stained for de novo synthesized RNA as per the legend to Figure 1 are shown. (**B**) Quantitative analysis for inhibition of de novo RNA synthesis (*n* = 20) as per the legend to Figure 1. (**C**) Results from (**B**) are expressed as percent inhibition of transcription relative to that effected by GFP-Mwt. Significant differences are indicated, * *p* < 0.01, *** *p* < 0.001.

**Figure 4 cells-10-02786-f004:**
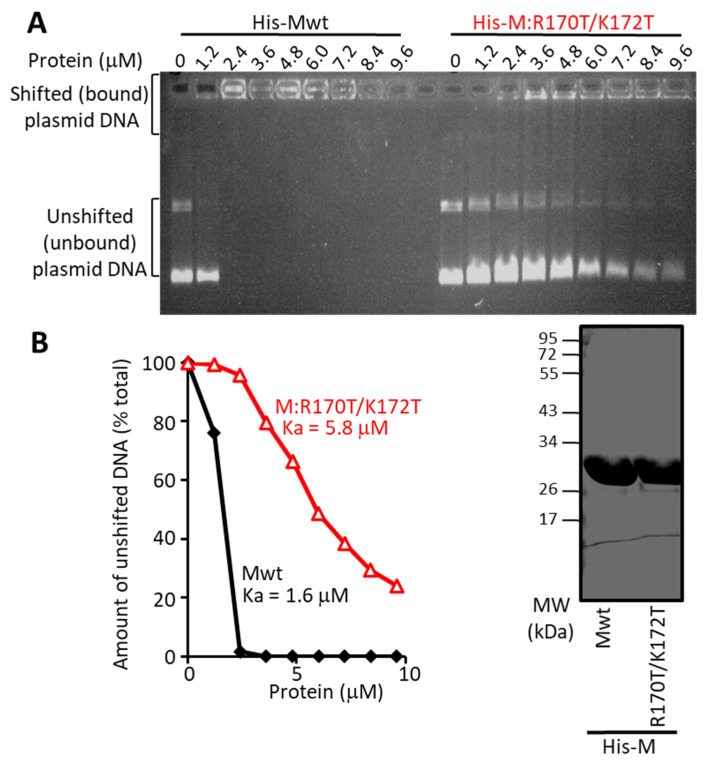
M can bind DNA in vitro dependent on residues R170 and K172. (**A**) Increasing concentrations of purified His-Mwt or His-M-R170T/K172T protein were incubated with 200 ng plasmid DNA (6.84 Kb) for 15 min at room temperature, and the protein-DNA complexes electrophoresed on an 0.8% agarose gel in the absence of ethidium bromide for 8 h at 4C. The gel was then stained with ethidium bromide solution (1 µg/mL) for 15 min, washed with water for 5 min and then imaged. (**B**) The levels of unshifted (unbound) DNA from (**A**) were quantified using the NIH ImageJ software, with results expressed as a percentage of the value in the absence of M protein. Association constants (Ka) estimated from the curves are indicated. Results are typical for a series of analogous experiments. Right: Coomassie blue stained SDS –PAGE gel of purified His-M proteins (4 µg/lane) used in (**A**). Running positions of molecular weight standards are shown on the left.

**Figure 5 cells-10-02786-f005:**
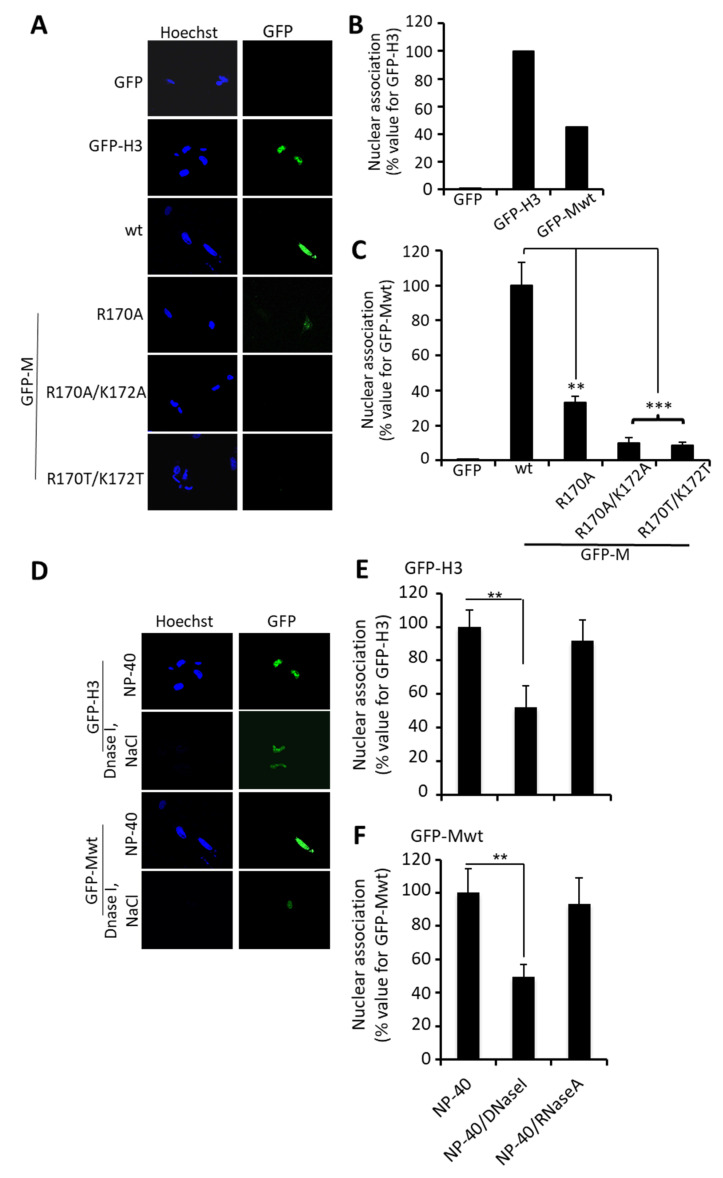
M can associate with chromatin in living cells dependent on residues R170 and K172. (**A**–**C**) Nuclear association of GFP alone or the indicated GFP fusion proteins (H3 histone, Mwt and mutants M-R170A and R170A/K172A) was assayed 24 h after transfection of HeLa cells by treating with 1% NP-40 for 15 min at room temperature. (**A**) Cells were subjected to CLSM; the green channel (GFP) is shown, with cellular dsDNA shown in blue (Hoechst). Note that not all cells in the panel are transfected. (**B**,**C**) Image analysis was performed using the NIH ImageJ software on CLSM files such as those shown (**A**) to determine the level of protein associated with the host cell chromatin. Results represent the mean +/− SEM (*n* = 15), expressed as a percentage relative to the value for GFP-H3 (**B**) or for the mutant derivatives of M relative to GFP-Mwt (**C**). (**D**–**F**). Cells were treated with NP40 as in (**A**) followed by incubation with DNase I (0.2 mg/mL) or RNase A (0.2 mg/mL) for 1 h at room temperature. (**D**) Cells were imaged as per (**A**). (**E**,**F**) Images such as those shown in (**D**) were analyzed as per (**B**,**C**). Results represent the mean +/− SEM (*n* = 15) as a percentage relative to nuclei treated with NP-40 alone, for GFP-H3 (**E**) and GFP-Mwt (**F**), respectively, as indicated. ** *p* < 0.01, *** *p* < 0.001.

**Figure 6 cells-10-02786-f006:**
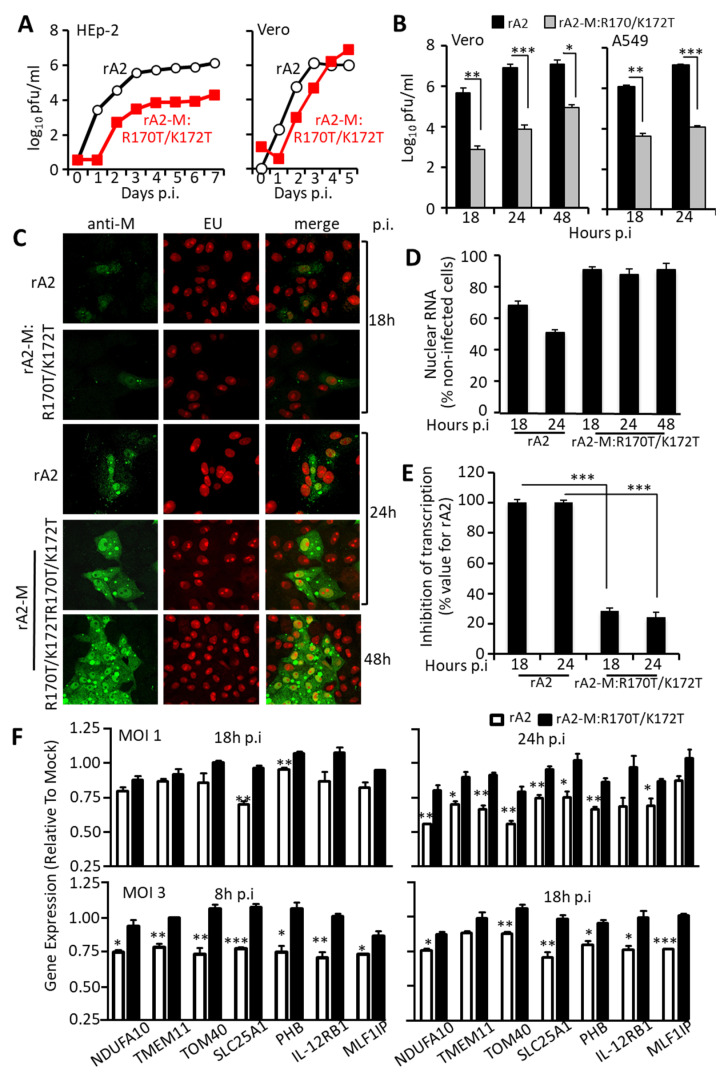
Recombinant RSV carrying R170T/K172T mutations in M shows impaired virus production and transcriptional inhibition. (**A**) HEp-2 (left) or Vero (right) cells were infected with recombinant wild type (rA2, open circles) or mutant (rA2-M:R170T/K172T, filled squares) virus at an MOI of 0.01, and samples collected every day for 7 (HEp-2) or 5 (Vero) days and virus titre estimated using immunostaining plaque assay. Results represent the mean +/− SEM (*n* = 3) for log plaque forming units (pfu)/mL. (**B**) Vero or A549 cells were infected with wild type rA2 (black columns) or mutant (rA2-M:R170T/K172T, grey columns) at an MOI of 3. Cells were harvested at the indicated times p.i. and analyzed for cell-associated infectious virus by plaque assays. Results represent the mean +/− SEM (*n* = 3) for log pfu/mL. * *p* < 0.05, ** *p* < 0.01, *** *p* < 0.001. (**C**) Vero cells were infected with rA2 or rA2-M:R170T/K172T at an MOI of 3 followed by incubation with 1 mM EU for 1 h at 18 or 24 h post-infection. Cells were then fixed, permeabilized, and specifically stained for de novo synthesized RNA and M protein, and CLSM imaged as per the legend to Figure 1. (**D**) Quantitative analysis for inhibition of de novo RNA synthesis (*n* = 20) as per the legend to Figure 1. (**E**) Results from (**D**) are plotted as the mean inhibition of nuclear transcription as a percentage relative to the value for rA2. *** *p* < 0.001. (**F**) A549 cells were infected without (mock) or with wild type rA2 or rA2-M:R170T/K172T at the indicated MOIs, prior to cell lysis at indicated times p.i. and RNA preparation as per the Methods section. Host gene expression was analyzed by qPCR (see Section 2.15), with expression levels relativized compared to gene expression levels in mock infected cells. Data represent the mean ± SEM (*n* = 3). * *p* < 0.05, ** *p* < 0.01, *** *p* < 0.001.

**Figure 7 cells-10-02786-f007:**
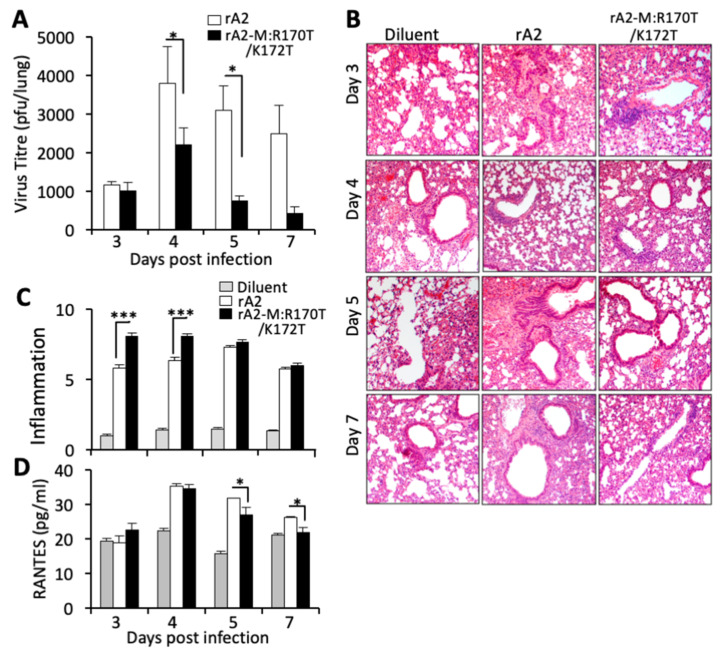
Recombinant RSV carrying R170T/K172T mutations in M induces higher inflammation and is cleared faster than wild type in a mouse model of RSV infection. BALB/c mice were inoculated with 50 µL of 2.5 × 10^5^ pfu of rA2 or rA2-R170T/K172T or an equivalent volume of diluent on day 0. Five mice from each group were euthanized on the indicated days p.i., and samples collected for analysis. (**A**) One lung from each animal was put into 1 mL of diluent with steel beads and frozen at 80 °C. Lungs were subsequently homogenized in a Tissue-lyser, debris removed by centrifugation and the supernatant used immediately for plaque assay. Results represent the mean +/− SEM (*n* = 5) for pfu/lung, representative of two separate experiments. White columns—rA2, black columns—rA2-R170T/K172T. * *p* < 0.05. (**B**) The second lung from each animal was fixed in formalin, embedded in paraffin, sectioned, and stained with haemotoxylin and eosin (H&E). Sections were imaged using a Leica DM750 microscope and the LAS software; five images were obtained at each of three different depths from each lung. Representative sections from each experimental group are shown, typical of two separate experiments. (**C**) Inflammation around bronchi was quantitated independently by two operators according to the schema described in Section 2.16 (0–9 scale [43]) using digital images such as those shown in (**A**). Quantitation was performed on multiple lung lobes from 3 different depths of sectioned tissue; results represent the mean + SEM (*n* ≥ 50). Grey columns—diluent, white columns—rA2, black columns—rA2-R170T/D172T. *** *p* < 0.001. (**D**) Blood was collected by cardiac puncture. Systemic inflammation was determined by ELISA for RANTES as described. Results represent the mean +/− SEM (*n* = 5). Grey columns—diluent, white columns—rA2, black columns—rA2-R170T/D172T. * *p* < 0.05.

**Figure 8 cells-10-02786-f008:**
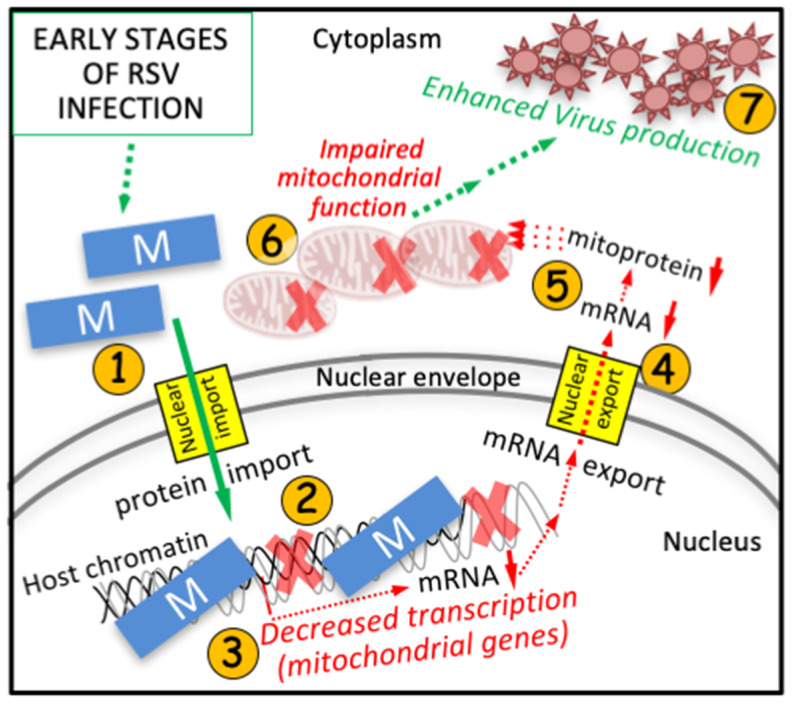
Novel mechanism by which RSV M inhibits host cell transcription through associating with cell chromatin. Key events early in RSV infection are shown, where host functions are indicated by red arrows, and events favoring viral infection are indicated by green arrows; dotted arrows denote multiple steps in the processes. After transport into the nucleus (**1**) [12], M associates with host cell chromatin (**2**) to suppress host cell transcription globally [49,51], with specific effects on nuclear-encoded mitochondrial genes (**3**). This results in reduced mRNA for mitochondrial gene products able to (**4**) be exported to the cytoplasm, leading to critical changes to the mitoproteome (**5**) [54,55], and severe impacts on host mitochondrial function (**6**) [50,56], which ultimately favor infectious virus production (**7**) [50,56]. Various agents that prevent RSV’s effects on the mitochondria reduce infectious virus production (not shown—see [50,56]. For early stages of RSV infection prior to M accessing the nucleus (not shown) see Hu et al. [51]; later in infection, M is exported to the cytoplasm to help coordinate nascent virion assembly (not shown) [4,6,7,8,9,14,51].

## Data Availability

All data are contained within the article and the Appendix A.

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
