# Peer review of "Respiratory Syncytial Virus Matrix Protein-Chromatin Association Is Key to Transcriptional Inhibition in Infected Cells"

_cells, 2021, doi:10.3390/cells10102786_

Round 1

Reviewer 1 Report

This manuscript found that Respiratory Syncytial Virus Matrix Protein-Chromatin Association is Key to Transcriptional Inhibition in Infected Cells. It is very interesting. One comment for discussion and figure 8. 

Please show the novelty in your study  and highlight your result in figure 8. 

Author Response

Rev 1

This manuscript found that Respiratory Syncytial Virus Matrix Protein-Chromatin Association is Key to Transcriptional Inhibition in Infected Cells. It is very interesting. One comment for discussion and figure 8. 

Please show the novelty in your study and highlight your result in figure 8. 

We thank the Reviewer for the very positive comments. We agree that we need to clarify the novel aspects of our study better, and above all integrate everything into Figure 8 in doing so. Accordingly, we have redrafted Figure 8 (new title “Novel mechanism by which RSV M inhibits host cell transcription through associating with host cell chromatin”) to much more clearly illustrate the mechanistic findings of the present study and its link to the findings of others. In addition, we have expanded the original Discussion paragraph into 2 paragraphs to much more clearly spell this out, with the new first paragraph in particular highlighting the new mechanistic information, and putting it in the context of existing literature and Figure 8. We thank the Reviewer for the most valuable suggestions, and encouraging us to highlight the novelty of our study. 

Reviewer 2 Report

Respiratory Syncytial Virus (RSV) causes inflation of small airways in the lungs.
Generally, the virus causes mild cold like symptoms, however it can cause complication in infants and elderly population.
This paper filled a gap in understanding the molecular mechanisms of RSV disease: the role of chromatin association matrix (M) inhibiting host cell transcription. The results evidence of the paper's hypothesis was confirmed in vitro and in vivo model. This important discover might implies in therapeutic development against RSV.
The manuscript is clearly written, the conclusion is justified and reference is updated. Therefore, I accept the paper without modifications.

Author Response

We thank the Reviewer for all of the positive comments.

Reviewer 3 Report

In Figure 1A, I would like to observe the infection free cells. I recommend to discuss the infection efficiency. I suggest to include brightfield photographs of cells and to argue why do you have only one cell when you stain with anti-M and you have several with EU.

In figure 2A I observed less signal in EU stain in 1-183 and 63-183 than in wt. I suggest you change the actual photograph for a different one that represents the graphic bar.  The photographs for 110-256 and 183-256 do not represent the graphic bars at least visually can you changed.

In figure 3. In photograph for K156T/K157T if you choose the upper cell for EU signal the result must be a complete transcription inhibition, how did you discern this issue? Please explain.

In photograph K172A I observed almost a complete transcription inhibition can you use another photograph or can you argue about this?

In figure 3A I observed less signal in EU for K130A than in Wt. Can you please explain?

Figure 4. The initial quantity of DNA in His-MWt and in His-M:R170T/k172T look different. In the second band of His-MWt where DNA and His-MWt interact with M protein the lower band diminishes, however, in the same for His-M:R170T/k172T an increase is observed, in general the DNA content for His-M:R170T/k172T seems more concentrated than in His-MWt. Another observation is that in 8.4 and 9.6 less DNA bound is observed versus 2.4, 3.6 and so on, actually the quantity is similar to 1.2 and the unbound DNA does not appear as expected. Can you explain this phenomenon? Or do you have another gel that represent the graph in figure 4C?.

In Figure 5 The second and third photographs of A and first and third of D are the same, can you add different photographs since you analyzed at least 15 cells. The photograph for RNase A treatment is not shown.

In Figure 6 M intensity seems higher in rA2 than in R170T/K172TR170T/K172T can you explain about this please?

In Figure 7 Can you describe and select with an arrow the damage caused by rA2 and R170T/K172T?

In general the results do not reflect the conclusions. 

Author Response

Rev 3

Reviewers 1 and 2 did not have any issues with any of the data of the paper, implying that Reviewer 3 may not understand our analysis using image analysis. It is important to appreciate the power of image analysis, which enables a quantitative estimation of the population of cells to be analysed, and thereby statistical analysis to determine significance; this is always a much better reflection of the overall result than a single contrast enhanced microscopic image without any quantitative analysis. Most importantly, as explained below, the rigour of the approach enables all manner of differences beyond experimental control, including cell to cell, image to image differences, can be corrected for to ensure that cell to cell variation is accounted for in every way.

  1. In Figure 1A, I would like to observe the infection free cells. I recommend to discuss the infection efficiency. I suggest to include brightfield photographs of cells and to argue why do you have only one cell when you stain with anti-M and you have several with EU.

The Reviewer seems to have misunderstood the point of how the data is presented in a user-friendly way to enable the way the analysis was done to be understood. Firstly, the conditions for RSV infection were chosen to ensure that infected and uninfected cells could be directly compared in the same image, i.e. not all cells are infected, as is quite clear from the staining (left column in A) – only in this way is it possible to compare infected and noninfected cells  next to one another for the effect in EU labelling. The analysis in B is precisely intended show the results for the specifically identified cells in A, showing the quantitative estimations for EU incorporation (Y axis) and nuclear accumulation (Fn/c numbers indicated), by way of illustration as to how the analysis is performed. The Reviewer has also not understood that of course multiple fields of cells (infected and non-infected) were analysed each time (not just the ones indicated in AB) and that ALL of the uninfected cell signals from the same image were quantitated and used to calculate the 100% EU incorporation; all of the individual infected cells were then compared individually cell by cell to this. The Reviewer will note that 15 cells in A are shown on the graph in C (not just 4). As the Reviewer will appreciate, analysis for the cell population is a much more reliable indicator of the result than a single contrast enhanced image that by definition cannot reflect the population variation.

To make all of this clearer for the Reviewer, we add additional explanation of our analyses to the legend for Figure 1C and F – this makes clear that analysis is for multiple cells, not just the cells shown in Figure 1A and D. We thank the Reviewer for encouraging us to clarify this point.

  1. In figure 2A I observed less signal in EU stain in 1-183 and 63-183 than in wt. I suggest you change the actual photograph for a different one that represents the graphic bar.  The photographs for 110-256 and 183-256 do not represent the graphic bars at least visually can you changed.

In contrast to Reviewers 1 and 2, the Reviewer appears not to have understood that a single cell image can never be representative of the whole population (hence why many more than one cell is always analysed in our work) and that the human eye is not able to quantitate differences in light very well (again, the reason for sophisticated softwares that enable this). Our analysis is performed in analogous fashion to Figure 1, with all non-transfected cells in this case quantified for EU and averaged to yield the 100% EU signal for the image, so that all transfected cells in the same image can be compared to it. This ensures that all sources of variations between images are corrected for to enable accurate estimation of the overall population of cells (the analysis for each construct in B/C is for 15 transfected cells, highlighting the mean +/- SEM; the statistical analysis demonstrates the significance). We thank the Reviewer for making us ensure our analysis is rigorous.

  1. In figure 3. In photograph for K156T/K157T if you choose the upper cell for EU signal the result must be a complete transcription inhibition, how did you discern this issue? Please explain.

In photograph K172A I observed almost a complete transcription inhibition can you use another photograph or can you argue about this?

The Reviewer is correct in terms of what he/she sees for a particular cell in the images, remembering the principle shown in the correlation shown in Figure 1D and F – more M in the nucleus results in more transcriptional inhibition. The Reviewer is reminded that the analysis in all of our work is for the population of cells – Figure 3A shows examples of some of the cells, while 3BC shows the quantitative analysis for 20 different cells for each construct, highlighting the mean +/- SEM; the statistical analysis demonstrates the significance. This applies for all of the Reviewers’ thoughts, with respect to the K156T/K157T and K172A constructs (as well as all of the others). We thank the Reviewer for making us ensure our analysis is rigorous.

  1. In figure 3A I observed less signal in EU for K130A than in Wt. Can you please explain?

The Reviewer is reminded that the analysis is performed in exactly the same way as for Figure 1/Figure 2 etc. with the untransfected cells in each image being averaged and used for the 100% for the transfected cells in the same image to be compared to. The data in 3BC is for n = 20 for K130A, as for all of the other constructs, with the SEM shown. We thank the Reviewer for making us ensure our analysis is rigorous.

  1. Figure 4. The initial quantity of DNA in His-MWt and in His-M:R170T/k172T look different. In the second band of His-MWt where DNA and His-MWt interact with M protein the lower band diminishes, however, in the same for His-M:R170T/k172T an increase is observed, in general the DNA content for His-M:R170T/k172T seems more concentrated than in His-MWt. Another observation is that in 8.4 and 9.6 less DNA bound is observed versus 2.4, 3.6 and so on, actually the quantity is similar to 1.2 and the unbound DNA does not appear as expected. Can you explain this phenomenon?

Reviewers 1 and 2 have not had this view. The Reviewer is reminded that unrestricted plasmid DNA is used in the experiment, which makes it relatively difficult to assess by eye without quantitation, and that the quantitative analysis relativises everything to the left lane as shown in the figure; i.e. the quantitative analysis is the key (as for Figs. 1-3). We now spell out that the result shown is typical of analysis from a series of analogous experiments. We thank the Reviewer for making us ensure our analysis is rigorous.

  1. In Figure 5 The second and third photographs of A and first and third of D are the same, can you add different photographs since you analyzed at least 15 cells.

We think the Reviewer means D and not C; we have revised the Figure as requested and thank him/her for pointing out this oversight.

  1. In Figure 6 M intensity seems higher in rA2 than in R170T/K172TR170T/K172T can you explain about this please?

Reviewers 1 and 2 did not have this view, and visual inspection (not quantitative, of course), indicates the intensity of green signal in cells infected with A2 or the mutant to be comparable. The Reviewer is reminded that our quantitative analysis is the key data, but we thank the Reviewer for making us ensure our analysis is rigorous.  

  1. In Figure 7 Can you describe and select with an arrow the damage caused by rA2 and R170T/K172T?

Unlike Reviewers 1 and 2, the Reviewer is confused – RSV does not cause direct cytopathic ‘damage’ in mouse lungs. As detailed in the text of the Results, the figure documents the effect of the significantly reduced viremia (Fig 7A) resulting from the mutant virus compared to wild type, and resulting increase in inflammation (Fig. 7BC) as assessed by thickening (increased number of cell layers) around the bronchii, and response in terms of inflammatory markers.

  1. In general the results do not reflect the conclusions. 

Reviewer 3 is alone in this view; Reviewers 1 and 2 are convinced by the data; the responses above should satisfy the Reviewer in all respects regarding the specific points above that relate to the importance of quantitative analysis of a population of cells, and not single contrast enhanced images.

We thank the Reviewer in making an important contribution to the rigorous review process, and identifying important point of confusion, which we have now cleared up in our revised manuscript. We are convinced the manuscript is a much better, and more rigorous and clearer evocation of our results, as a result of his/her comments, together with those of Reviewers 1 and 2.